# Current Status of Mumps Virus Infection: Epidemiology, Pathogenesis, and Vaccine

**DOI:** 10.3390/ijerph17051686

**Published:** 2020-03-05

**Authors:** Shih-Bin Su, Hsiao-Liang Chang, Kow-Tong Chen

**Affiliations:** 1Department of Occupational Medicine, Chi-Mei Medical Center, Tainan 710, Taiwan; shihbin.su@msa.hinet.net; 2Department of Surveillance, Centers for Disease Control, Taipei 100, Taiwan; hlchang@cdc.gov.tw; 3Department of Occupational Medicine, Tainan Municipal Hospital (managed by Show Chwan Medical Care Corporation), Tainan 701, Taiwan; 4Department of Public Health, College of Medicine, National Cheng Kung University, Tainan 701, Taiwan

**Keywords:** mumps, epidemiology, pathogenesis, vaccine

## Abstract

Mumps is an important childhood infectious disease caused by mumps virus (MuV). We reviewed the epidemiology, pathogenesis, and vaccine development of mumps. Previous studies were identified using the key words “mumps” and “epidemiology”, “pathogenesis” or “vaccine” in MEDLINE, PubMed, Embase, Web of Science, and Google Scholar. We excluded the articles that were not published in the English language, manuscripts without abstracts, and opinion articles from the review. The number of cases caused by MuV decreased steeply after the introduction of the mumps vaccine worldwide. In recent years, a global resurgence of mumps cases in developed countries and cases of aseptic meningitis caused by some mumps vaccine strains have renewed the importance of MuV infection worldwide. The performance of mumps vaccination has become an important issue for controlling mumps infections. Vaccine development and routine vaccination are still effective measures to globally reduce the incidence of mumps infections. During outbreaks, a third of MMR vaccine is recommended for groups of persons determined by public authorities.

## 1. Introduction

Mumps is known as an important vaccine-preventable childhood viral disease [1]. The clinical pictures of mumps virus (MuV) infection is characterized by pain and swelling of the parotid glands, but can involve various other tissues and organs [2]. It can cause serious complications including encephalitis, meningitis, orchitis, myocarditis, pancreatitis, and nephritis [2,3]. Although mumps is a benign disease, often with complete recovery within a few weeks after being infected, long-term outcomes, such as seizures, cranial nerve palsies, hydrocephalus, and deafness, can occur [3,4]. Due to its benign clinical picture, mumps has been somewhat neglected compared with other infectious diseases (e.g., measles). However, in 2016 and 2017, the number of cases of mumps increased almost two-times compared to those reported in the previous five years in the United States (US) [5]. Additionally, in recent years, several huge outbreaks of the mumps infection have been reported in developed countries [6]. How to prevent and control the epidemic of mumps has sparked a new problem of public health. Previous report showed that 70% of patients with known vaccination history had received two doses of measles, mumps, and rubella (MMR) vaccine prior to illness [5]. Compared to mumps, cases of measles and rubella rarely occur in person with two doses of MMR vaccine, suggesting some factors of mumps vaccine influences its effectiveness [3]. Therefore, understanding the characteristics of MuV epidemiology, pathogenesis, and vaccine is important to protect against the spread of MuV infection. The purpose of this study was to summarize the epidemiology, pathogenesis, and vaccine of mumps and recommendations arising from this project.

All papers used in this review were published 2002 or later. These papers related to mumps virus infection were obtained by searching online database of MEDLINE (National Library of Medicine, Bethesda, MD, USA), PubMed, Embase, Web of Science, and Google Scholar using in combination of key terms “mumps” and “epidemiology”, “pathogenesis”, or “vaccine”. The articles for review were restricted to articles published in the English language. Additional sources were identified from the references of relevant literature. Articles without abstracts and opinion articles were excluded from the review. After selecting the articles, potential information related to epidemiology, pathogenesis, and vaccine of mumps were extracted and classified according to the categories of epidemiology, pathogenesis, and vaccines.

Searches of literatures were carried out in September and October 2019. A total of 3002 papers were identified from the study sources and this was reduced to 124 after screening the un-relevant articles and duplicates were removed (Figure 1).

## 2. Epidemiology

Mumps was a serious disease that cause significant morbidity and mortality worldwide before the program of mumps vaccination starting [5]. In the pre-vaccine era, mumps was a severe contagious disease with a high morbidity of approximately 40–726 cases per 100,000 population per year [7,8,9,10,11,12,13,14,15,16,17,18,19,20,21,22,23,24] (Table 1). During the pre-vaccine era, mumps was circulating endemically with a periodic spike of two to five years and a peak incidence of infection among children aged five to seven years old in several regions globally [23,24,25,26]. Mump infections frequently occurred in crowded population centers, e.g., prisons, kindergartens, boarding schools, military barracks, and other similar crowded settings [22]. Several risk factors have been reported in mumps infection including age, exposure, compromised immunity, time of year, travel, and vaccination status. Although there is no evidence of a difference on occurrence of MuV infections between the sexes, males seemly have higher risk to present complications [22,23]. Previous serosurvey studies were conducted in several countries prior to the introduction of vaccines [21,27,28,29]. They showed that 50% of children aged four to six years old and 90% of children aged 14–15 years old were seropositive, with a steep increase in mumps antibody levels at two to three years of age. These results indicate that almost all individuals without receiving mumps vaccination will eventually become infected [21].

In the vaccine era, the widespread use of mumps vaccines has substantially reduced the risk of occurrence as well as the number of serious complications due to mumps [1,20]. The rate of mumps infection was greatly reduced after the introduction of mumps vaccination (Table 1). The epidemiologic pattern of mumps changing is based on the various vaccination program, such as the number of doses of vaccination, age at vaccination, and vaccine coverage. The coverage rate of mumps vaccination increases, the average age at infection increases until the level of seropositive population needed to stop spread of mumps has been achieved [29,30]. If vaccine coverage rate is insufficient, it can lead to an increase in the burden of severe sequelae as the disease shifts to older age groups in which mumps complications are more prevalent [21,31,32,33,34].

A variation in the peak season of mumps infections between different years has been reported in some studies. In temperate climate, the occurrences of MuV infections are strong seasonality, with peaks in winter and spring [25,35]. A higher risk for mumps infections was observed during the summer months than during other seasons in Asia, and epidemics recur with a seasonal pattern [35,36,37]. A previous study showed that the occurrence of MuV infection increased as temperature and humidity increased [35]. The reasons of the seasonal pattern of mumps infection may be due to the following factors: the fluctuations of human immune competence mediated by seasonal factors, such as levels of melatonin [38]; seasonality-related behavioral factors, such as school attendance and indoor crowding [26]; and meteorological factors [36,37,39], including temperature, sunshine, wind, and relative humidity. However, human behavioral factors alone do not explain the seasonal pattern observed for certain cases of mumps infection [38,40].

Although mumps was initially successfully controlled by vaccination in developed countries [31,32,33,34,41], sporadic mumps outbreaks began to occur globally [42,43,44,45,46,47,48,49]. During recent years, outbreaks of MuV infections have occurred in adolescent populations, many of whom had been vaccinated with mumps vaccines previously, in the USA [46,50], Canada [42], Australia [49,51], United Kingdom [34,48], and France [14,47]. Several reasons were raised to explain the unexpected occurrence: waning immunity [52,53]; the efficacy of mumps vaccine, which has varied according to the doses of vaccinations and different virus strain used for production of the mumps vaccine [54,55,56,57]; and how much the level of antibody persisted in body with time after vaccination or natural infection among population [24,43].

## 3. Pathogenesis

Both mumps and parainfluenza virus belong to the paramyxovirus group and are antigenically related. It has been reported that parainfluenza virus has been isolated from cases of parotitis infection [39] and MuV has been isolated from patients with upper respiratory infection without inflammation of salivary gland [58]. Paramyxoviridae is an enveloped particle containing a non-segmented negative strand RNA molecule of 15,384 nucleotides [40]. The mumps virus is antigenically stable and does not exhibit great shifts in ways such as influenza virus. The incubation period for mumps ranges from 12–25 days post exposure and parotitis typically occurs 16–18 days after exposure [2]. There are at least 12 genotypes which are defined based on the sequence of the small hydrophobic (SH) gene [59]. Currently, the most frequently detected mumps genotype in recent outbreaks worldwide is genotype G [60].

MuV is transmitted to people by respiratory or oral route with infected respiratory droplets or secretions and the incubation time ranges from two to four weeks [61]. The virus has been isolated from saliva from seven days before until eight days after the occurrence of clinical symptoms [62]. Following exposure, MuV infects the upper respiratory tract via binding to sialic acid to enter polarized epithelial cells in the respiratory tract and enhance the invasion of mumps virus to neighboring cells [63]. MuV is markedly secreted from the epithelial cells, which causes virus growth in the glandular epithelium and shedding of mumps virus in saliva [63]. MuV can spread systemically in human body resulting in viremia during the early phase of infection [64]. Humans are known as the only natural host of MuV. Most (approximately one-half) cases are asymptomatic or suffer from only mild respiratory symptoms or fever after MuV infections [65]. Classic mumps infection is characterized by parotitis, but inflammation of salivary gland is not a main or necessary clinical manifestation of mumps infection [66]. Other organs, including the central nervous system (CNS) [67], heart, kidney [68], and genital organs [69] can also be affected via viremia dissemination. Viremia seems to be inhibited by the humoral antibodies and the virus level in salivary secretion correlates inversely with the local level of virus-specific secretory IgA produced [70]. It was hypothesized that neutralizing antibody that is produced in salivary glands may play an important role in restricting the replication of mumps virus and its excretion into saliva. In addition, the level of T cell antibody may play a role in the inhibition and clearance of mumps virus. It has been assumed that mumps virus invades T cells and efficiently grows in these cells [71]. Migrating mumps virus-infected T cells could enhance the mumps virus to disseminate to the various organs and might therefore play a key role in the development of mumps disease [71].

## 4. Mumps Vaccines

### 4.1. General Considerations

The first inactivated mumps vaccine was developed and used in the USA in 1950 [1]. Until the 1960s, live attenuated mumps virus vaccines were first used in the USA and Soviet Union [2,72]. They used different virus strains to develop mumps vaccines. The USA used the Jerry Lynn vaccine strain, whereas the Soviet Union used a different strain called the Leningrad-3 strain. Additionally, several other virus strains, such as Rubini, Urabe, RIT 4385, Leningrad-Zagreb (L-Z), Miyahara, Hoshino, Tori, NK M-46, and S-12, have been used worldwide to produce mumps vaccines [59,72].

The Jeryl Lynn and Urabe Am9 strains have been the most popularly used in the world, followed by the Leningrad-Zagreb, Leningrad-3, and Rubini strains. The newer RIT 4385 strain was derived from the Jeryl Lynn strain [21,72,73,74,75]. Initially, mumps vaccines (panel) were produced as a form of monovalent vaccine, and mumps vaccines are currently manufactured as a component of the trivalent measles-mumps-rubella (MMR) vaccine worldwide [73]. At least five known combination MMR vaccines have been developed since the 1960s worldwide (Table 2) [72,73,74,75,76]. To date, the MMR vaccine has been used in more than 120 countries worldwide [2,76]. Since the mid-2000s, there are two measles-mumps-rubella-varicella (MMRV) vaccines available: ProQuad (Merck and Co. Inc. West Point, PA, USA), and Priorix-Tetra (GlaxoSmithKline Biologicals, Rixensart, Belgium) (Table 3) [76,77,78]. Vaccination against MMR or MMRV has recently been conducted mainly with combination vaccines.

### 4.2. Jeryl Lynn Strain Mumps Vaccine

The Jeryl Lynn strain, named after a woman who developed mumps with unilateral parotitis in 1963, was the first vaccine strain to be produced in the USA by passaging the virus in embryonated hen eggs and chicken embryo fibroblast cultures [76,79,80,81]. The strain was licensed in the USA in 1967 and since then a live attenuated Jeryl Lynn vaccine has been used in many countries [82]. Another mumps vaccine, the RIT 4385 strain, was developed from a single Jeryl Lynn clone by passage through chick embryo fibroblast cultures [83]. A clinical study conducted in an industrialized country found that the seroconversion rates after vaccination with a single dose of the Jeryl Lynn strain mumps vaccine were 80%–100% [84]. Several studies have presented existence of antibodies in a high ratio of vaccinated populations. In Sweden, 73% (*n* = 167) children who vaccinated with one dose of MMR vaccine (containing the Jeryl Lynn strain) at 18 months of age persisted to be seropositive at time of vaccination 12 years later, and 93% were seropositive after a second dose of mumps vaccination [85]. In Finland [86], a 9-year serological follow-up period was performed among 254 children. Among these children, one dose of MMR vaccine (containing the Jeryl Lynn mumps strain) was given to children at the ages of 14–18 months and six years. The seroconversion rate was 86% among the vaccinated children, and the seropositivity rate was 95% after revaccination. These results indicated that a two-dose MMR vaccination resulted in a high mumps immunity level. In the Dominican Republic [87], a clinical evaluation of a new measles (Schwarz strain), mumps (Jeryl Lynn strain), and rubella (Cendehill strain) combined live vaccine was conducted. In this study, there were more than 900 study children with a 94% seroconversion rate for mumps vaccines among the study children aged one to six years [87]. It was found that the occurrence of mumps increased with time elapse since vaccination in the epidemic of mumps in USA, suggesting waning of immunity is likely to play a role [88,89].

The Jeryl Lynn strain, named after a woman who developed mumps with unilateral parotitis in 1963, was the first vaccine strain to be produced in the USA by passaging the virus in embryonated hen eggs and chicken embryo fibroblast cultures [76,79,80,81]. The strain was licensed in the USA in 1967 and since then a live attenuated Jeryl Lynn vaccine has been used in many countries [82]. Another mumps vaccine, the RIT 4385 strain, was developed from a single Jeryl Lynn clone by passage through chick embryo fibroblast cultures [83]. A clinical study conducted in an industrialized country found that the seroconversion rates after vaccination with a single dose of the Jeryl Lynn strain mumps vaccine were 80%–100% [84]. Several studies have presented existence of antibodies in a high ratio of vaccinated populations. In Sweden, 73% (*n* = 167) children who vaccinated with one dose of MMR vaccine (containing the Jeryl Lynn strain) at 18 months of age persisted to be seropositive at time of vaccination 12 years later, and 93% were seropositive after a second dose of mumps vaccination [85]. In Finland [86], a 9-year serological follow-up period was performed among 254 children. Among these children, one dose of MMR vaccine (containing the Jeryl Lynn mumps strain) was given to children at the ages of 14–18 months and six years. The seroconversion rate was 86% among the vaccinated children, and the seropositivity rate was 95% after revaccination. These results indicated that a two-dose MMR vaccination resulted in a high mumps immunity level. In the Dominican Republic [87], a clinical evaluation of a new measles (Schwarz strain), mumps (Jeryl Lynn strain), and rubella (Cendehill strain) combined live vaccine was conducted. In this study, there were more than 900 study children with a 94% seroconversion rate for mumps vaccines among the study children aged one to six years [87]. It was found that the occurrence of mumps increased with time elapse since vaccination in the epidemic of mumps in USA, suggesting waning of immunity is likely to play a role [88,89].

The Jeryl Lynn strain, named after a woman who developed mumps with unilateral parotitis in 1963, was the first vaccine strain to be produced in the USA by passaging the virus in embryonated hen eggs and chicken embryo fibroblast cultures [76,79,80,81]. The strain was licensed in the USA in 1967 and since then a live attenuated Jeryl Lynn vaccine has been used in many countries [82]. Another mumps vaccine, the RIT 4385 strain, was developed from a single Jeryl Lynn clone by passage through chick embryo fibroblast cultures [83]. A clinical study conducted in an industrialized country found that the seroconversion rates after vaccination with a single dose of the Jeryl Lynn strain mumps vaccine were 80%–100% [84]. Several studies have presented existence of antibodies in a high ratio of vaccinated populations. In Sweden, 73% (*n* = 167) children who vaccinated with one dose of MMR vaccine (containing the Jeryl Lynn strain) at 18 months of age persisted to be seropositive at time of vaccination 12 years later, and 93% were seropositive after a second dose of mumps vaccination [85]. In Finland [86], a 9-year serological follow-up period was performed among 254 children. Among these children, one dose of MMR vaccine (containing the Jeryl Lynn mumps strain) was given to children at the ages of 14–18 months and six years. The seroconversion rate was 86% among the vaccinated children, and the seropositivity rate was 95% after revaccination. These results indicated that a two-dose MMR vaccination resulted in a high mumps immunity level. In the Dominican Republic [87], a clinical evaluation of a new measles (Schwarz strain), mumps (Jeryl Lynn strain), and rubella (Cendehill strain) combined live vaccine was conducted. In this study, there were more than 900 study children with a 94% seroconversion rate for mumps vaccines among the study children aged one to six years [87]. It was found that the occurrence of mumps increased with time elapse since vaccination in the epidemic of mumps in USA, suggesting waning of immunity is likely to play a role [88,89].

The Jeryl Lynn strain, named after a woman who developed mumps with unilateral parotitis in 1963, was the first vaccine strain to be produced in the USA by passaging the virus in embryonated hen eggs and chicken embryo fibroblast cultures [76,79,80,81]. The strain was licensed in the USA in 1967 and since then a live attenuated Jeryl Lynn vaccine has been used in many countries [82]. Another mumps vaccine, the RIT 4385 strain, was developed from a single Jeryl Lynn clone by passage through chick embryo fibroblast cultures [83]. A clinical study conducted in an industrialized country found that the seroconversion rates after vaccination with a single dose of the Jeryl Lynn strain mumps vaccine were 80%–100% [84]. Several studies have presented existence of antibodies in a high ratio of vaccinated populations. In Sweden, 73% (*n* = 167) children who vaccinated with one dose of MMR vaccine (containing the Jeryl Lynn strain) at 18 months of age persisted to be seropositive at time of vaccination 12 years later, and 93% were seropositive after a second dose of mumps vaccination [85]. In Finland [86], a 9-year serological follow-up period was performed among 254 children. Among these children, one dose of MMR vaccine (containing the Jeryl Lynn mumps strain) was given to children at the ages of 14–18 months and six years. The seroconversion rate was 86% among the vaccinated children, and the seropositivity rate was 95% after revaccination. These results indicated that a two-dose MMR vaccination resulted in a high mumps immunity level. In the Dominican Republic [87], a clinical evaluation of a new measles (Schwarz strain), mumps (Jeryl Lynn strain), and rubella (Cendehill strain) combined live vaccine was conducted. In this study, there were more than 900 study children with a 94% seroconversion rate for mumps vaccines among the study children aged one to six years [87]. It was found that the occurrence of mumps increased with time elapse since vaccination in the epidemic of mumps in USA, suggesting waning of immunity is likely to play a role [88,89].

The Jeryl Lynn strain, named after a woman who developed mumps with unilateral parotitis in 1963, was the first vaccine strain to be produced in the USA by passaging the virus in embryonated hen eggs and chicken embryo fibroblast cultures [76,79,80,81]. The strain was licensed in the USA in 1967 and since then a live attenuated Jeryl Lynn vaccine has been used in many countries [82]. Another mumps vaccine, the RIT 4385 strain, was developed from a single Jeryl Lynn clone by passage through chick embryo fibroblast cultures [83]. A clinical study conducted in an industrialized country found that the seroconversion rates after vaccination with a single dose of the Jeryl Lynn strain mumps vaccine were 80%–100% [84]. Several studies have presented existence of antibodies in a high ratio of vaccinated populations. In Sweden, 73% (*n* = 167) children who vaccinated with one dose of MMR vaccine (containing the Jeryl Lynn strain) at 18 months of age persisted to be seropositive at time of vaccination 12 years later, and 93% were seropositive after a second dose of mumps vaccination [85]. In Finland [86], a 9-year serological follow-up period was performed among 254 children. Among these children, one dose of MMR vaccine (containing the Jeryl Lynn mumps strain) was given to children at the ages of 14–18 months and six years. The seroconversion rate was 86% among the vaccinated children, and the seropositivity rate was 95% after revaccination. These results indicated that a two-dose MMR vaccination resulted in a high mumps immunity level. In the Dominican Republic [87], a clinical evaluation of a new measles (Schwarz strain), mumps (Jeryl Lynn strain), and rubella (Cendehill strain) combined live vaccine was conducted. In this study, there were more than 900 study children with a 94% seroconversion rate for mumps vaccines among the study children aged one to six years [87]. It was found that the occurrence of mumps increased with time elapse since vaccination in the epidemic of mumps in USA, suggesting waning of immunity is likely to play a role [88,89].

Natural mumps infection has been considered to give life-long immunity. However, the duration of vaccine-induced immunity has not been well documented, but detectable specific mumps virus-neutralizing antibodies correlate with resistance to clinical infection [90]. It has been found that the efficacy of the mumps vaccine containing the Jeryl Lynn strain was 95–96% [90]. The effectiveness of two doses of MMR immunization is not as high as expected, and outbreaks have been reported in countries with high vaccine coverage [91,92,93]. It was reported that the estimated vaccine effectiveness for one-dose MMR vaccination was 85.4% (95% CI: 67.3–93.4%), and that for two-dose MMR vaccination was 88.5% (95% CI: 78.1–93.9%) in a study in Spain [94].

There was a study with a 10-year retrospective study of hospitalized cases of mumps in the USA that did not find an increased risk of aseptic meningitis after MMR containing the Jeryl Lynn strain of mumps (odds ratio <1.0 for all analyses) [95].

Since the mid-2000s, measles-mumps-rubella-varicella (MMRV) vaccines have been available [96]. The MMRV vaccine was developed based on the existing MMR and varicella vaccines. Licensed ProQuad (Merck and Co., Inc. West Point, PA, USA; Merk) and Priorix-Tetra (GlaxoSmithKline Biologicals, Rixensart, Belgium; GSK) have different measles virus strains (Edmonston strain and Schwarz strain, respectively), different mumps virus strains (RIT 4385 strain and Jeryl Lynn strain, respectively), the same rubella virus strain (Wistar RA 27/3 strain), and the same varicella virus strain (Oka strain) [97]. The RIT 4385 strain was developed by cloning a major population of the Jeryl Lynn strain [98,99]. Similar seroconversion rates of these four viruses were found between comparison groups [59,79]. Recent evidence has shown that the MMRV vaccines have comparable immunogenicity and safety profiles in healthy children [97]. The Jeryl Lynn mumps genotype A strain has been used in the MMR vaccine throughout the world. It shows a markedly lower incidence of aseptic meningitis than other strains and higher immune response [79].

### 4.3. Leningrad-3 Strain Mumps Vaccine

The Leningrad-3 mumps attenuated strain was produced in the Soviet Union using the various strains obtained after 16 serial passages in guinea pig kidney tissue culture, and then further passages in Japanese quail embryo cultures [100]. Since 1974, vaccines (containing Leningrad-3 strain) have been used in the former Soviet Union. It is estimated that 8–11 million doses of the Leningrad-3 mumps vaccine is produced every year [78]. There was a study to explore the seroconversion rates after vaccination with an MMR vaccine (measles vaccine with the Leningrad-16 strain, mumps vaccine with the Leningrad-3 strain, and rubella virus vaccine with the Leningrad-8 strain) [100]. This study showed that the mumps seroconversion rate was 89%–98% and that there was a protective efficacy of 92%–99% after vaccination with the MMR vaccine among the study children aged between one and seven years [100]. During 1979–1986, there was a retrospective study to review medical records in the Slovenian population [101]. It was found the occurrence of aseptic meningitis was 100 cases per 100,000 doses of measles-mumps vaccine (containing the Leningrad-3 mumps strain), however, all patients with aseptic meningitis recovered with no sequelae [101].

### 4.4. Leningrad-3-Zagreb Strain Mumps Vaccine

In Croatia and India, the Leningrad-3-Zagreb (L-Zagreb) strain was developed by further passage of the Leningrad-3 mumps virus by adaptation in chick embryo fibroblast cell culture [79,102]. The L-Zagreb strain has been used to prepare mumps vaccines. Between 1976 and 1987, it was estimated that over 10 million doses of the L-Zagreb mumps vaccine were used in the former Yugoslavia and elsewhere [102]. A study showed that the positive rate of IgG levels four weeks after immunization rose from 12% to 92% for mumps, with a vaccine efficacy of 97%–100% [103]. Only mild side effects, including pain and swelling in 37 (4.3%) cases, mild fever in 51 (5.9%) cases, cough in 40 (4.6%) cases, and a transient rash in seven (0.8%) cases, were observed [103]. In Croatia, there were 90 cases of aseptic meningitis per 100,000 doses of MMR vaccine containing the L-Zagreb mumps strain between 1988 and 1992 [104]; this finding is similar to the finding reported by Cizman et al. (10 per 10,000) [101].

### 4.5. Rubini Strain Mumps Vaccine

The Rubini strain was derived from a child of the same name presenting typical clinical signs and symptoms of mumps infection. A live attenuated mumps vaccine virus strain for human diploid cells was developed at the Swiss Serum and Vaccine Institute, Berne [105]. Attenuation of the wild virus was performed by isolation and serial passage in WI-38 human diploid cells, specific pathogen-free hen eggs and MRC-5 human diploid cells [105]. A mumps vaccine containing the Rubini strain was approved and commercialized in Switzerland in 1985, and over four million people worldwide had received this vaccine by 1990, [74].

In Germany, children aged 14–24 months who received one dose of the MMR vaccine was conducted to explore the immunity response [106]. This study found that the seroconversion rate was 95% among children who received the Triviraten vaccine (containing the Rubini mumps strain), whereas the seroconversion rate was 100% when the mumps strain was Jeryl Lynn [106]. In Switzerland, a study was conducted to explore the secondary attack rates among family contacts and confirmed mumps case contacts [106]. It found a protective efficacy among persons vaccinated with mumps vaccine (containing the Rubini strain) was 6%, whereas the protective efficacy among persons vaccinated with mumps vaccine (containing the Urabe strain) was of 73%, and the persons vaccinated with mumps vaccine (containing Jeryl Lynn strain) was 62% [107]. In Italy, a case–control study was performed between 1995 and 1996 [108]. This study showed the children vaccinated with the Rubini strain mumps vaccine had a higher risk of MuV infection, compared with children vaccinated with a Jeryl Lynn or Urabe strain mumps vaccine [108]. A case-cohort study comparing the effectiveness of vaccination with the Rubini strain to vaccination with the Jeryl Lynn strain vaccine was performed in Switzerland [109]. The effectiveness of vaccination with the Rubini strain was zero and that with the Jeryl Lynn strain was 70% against clinical mumps [109].

### 4.6. Urabe Strain Mumps Vaccine

The Biken Institute in Japan developed the Urabe Am9 strain vaccine from mumps virus isolated from the saliva of a patient in 1979, and thereafter, the vaccine was licensed in Belgium, France, Italy, and many countries worldwide [76,78].

The Urabe Am9 strain was evaluated in children 14 to 20 months of age in a comparative trial with the Jeryl Lynn strain [106]. The seroconversion rates at six weeks, as detected with either one or two tests, were 94.8% after the Urabe Am9 strain and 96.7% after the Jeryl Lynn vaccine. Only mild infrequent adverse reactions were observed. Both strains of live attenuated mumps vaccine were immunogenic and well tolerated in this age group.

There was a study in Finland [106], in which the children who received a mumps vaccine at 14–20 months of age. This study found the seroconversion rate after vaccination with a Urabe strain vaccine was 95%, compared with seroconversion rate of 97% among children vaccinated with the Jeryl Lynn strain [106]. According to the information obtaining from an outbreak investigation of mumps in an Asian population in Singapore [107], the vaccine efficacies of the Jeryl Lynn strain, Urabe strain and Rubini strain mumps vaccines were 81%, 54%, and 55%, respectively.

The immune response after vaccination with the mumps vaccine (containing Urabe strain) has been studied in developing countries. A lower antigen response was found in subjects with measles and mumps vaccination for a nine-month-old group than for a 12–15-month-old group in India [110]. Among children aged 12 months, the immunogenicity response rate was 98% in Taiwan [111] and 92% in India [110]. In the United Kingdom, a study showed that the seropositivity rates were 85% at 4 years after a single dose of MMR vaccination with the Urabe strain vaccine, compared with 81% for vaccination with the Jeryl Lynn strain [112]. In Canada, a study found that five to six years after one dose of MMR vaccine, the seropositivity rate was 93% for the Urabe strain, compared with 85% for the Jeryl Lynn strain [113,114,115].

In 1997, a study on children aged 1–11 years vaccinated with MMR (containing the Urabe strain) vaccine was conducted in northeastern Brazil [116]. This study showed a higher occurrence of aseptic meningitis 3 weeks after Brazil’s national vaccination day (relative risk = 14.3; 95% confidence interval: 7.9, 25.7), compared with the risk in the pre-vaccination period. This result was confirmed by a case series analysis (relative risk = 30.4; 95% confidence interval: 11.5, 80.8). The estimated risk of aseptic meningitis among children who received MMR (containing Urabe strain) vaccination was 1 in 14,000 doses [116].

Canada developed molecular studies on mumps vaccine when the authorities received the reports of aseptic meningitis cases related to the vaccination with MMR vaccine (containing the Urabe mumps virus strain) [117]. It was found that the Urabe vaccine was a mixture wild type A and variant G virus strains [116]. The MMR vaccine (containing the Urabe strain) was therefore withdrawn from the market in Canada in 1990 [118]. The reason for the high rate of vaccine-associated disease with the Urabe AM9 vaccine is not clear [114]. A study in United Kingdom showed that the rate of aseptic meningitis was 9 cases per 100,000 vaccine doses [119]. This result was confirmed by a study conducted in multicenter. According to these findings, mumps vaccine (containing Urabe strain) was stopped purchasing in the United Kingdom in 1992. In Japan, nationwide surveillance was run by the Japan Ministry of Health and Welfare during 1989, and reported an incidence of 49 cases of aseptic meningitis per 100,000 doses of MMR vaccine (containing the Urabe mumps strain) [120]. Until 1993, a cumulative incidence of aseptic meningitis achieved 100 cases per 100,000 doses of MMR (containing the Urabe mumps strain), therefore all MMR vaccines were withdrawn from market of Japan in April 1993 [121].

## 5. Emerging Mumps Virus Infection

Whereas most (>90%) of children experienced mumps infection by age 20 in pre-vaccine era, number of mumps cases declined sharply after introduction of mumps vaccine globally (Table 1). Incidence of MuV infection among vaccinated school- and high school-age children increased in the late 1980s, followed by sustained decrease in incidence after children were recommended to receive a second dose of MMR vaccine at four six years of age [41]. However, resurgence of outbreaks of MuV infection continues to occur in developed countries in recent years [7,18,29]. Based on the previous studies [77,83,85], the vaccine has proven itself to be highly efficacious. What factors contribute to the occurrence of mumps cases? There are several hypotheses to explain the increasing number of mumps outbreak: (1) antigenic differences between 1967 vaccine strain and contemporary circulating strains might permit immune escape. However, this theory seems unlikely given that sera collected from individuals shortly after vaccination have been shown to effectively neutralize a mass array of genetically disparate virus strains [122]; (2) waning of vaccine-induced immunity have been proposed as a likely contributor of mumps outbreaks [88]. Several studies have shown time after vaccination with declining concentrations of mumps virus-specific antibodies, decreased vaccine effectiveness, and increased risk of suffering from mumps virus infection [7,123].

The hypothesis that waning immunity is a cause of global resurgence in mumps cases suggests administration of an additional dose of vaccine during adolescence [77,124]. Between 2016 and 2017, there are numerous mumps outbreaks occurred throughout United States. To reflex these urgent events, the Advisory Committee on Immunization Practices (ACIP) recommended a third dose of MMR vaccine for groups of persons justified by public health authorities at increased risk for acquiring mumps due to an outbreak of mumps [5]. Subsequently, a guidance was developed by the Centers for Disease Control (CDC) and Prevention in the US to assist public health authorities when making decisions that effect groups at risk of acquiring mumps. Such groups should receive a third dose of the mumps vaccine. Based on the CDC guidance, public health authorities may opt to use when considering a third dose of MMR for controlling mumps outbreak [5].

## 6. Conclusions

In summary, although mumps is a benign clinical disease, it has become an important re-emerging pathogen. The reemergence of MuV infection will continue to threaten people unless additional preventive measures are implemented. Mumps vaccine strains have been used in the world and have shown that live mumps vaccines are highly cost-effectiveness for vaccination, despite the occasional occurrence of aseptic meningitis. Improving our understanding of mumps virus infection epidemiology, pathogenesis, and concerns over vaccine safety and efficacy are important steps to better intervene in the mumps virus infection spread.

## Figures and Tables

**Figure 1 ijerph-17-01686-f001:**
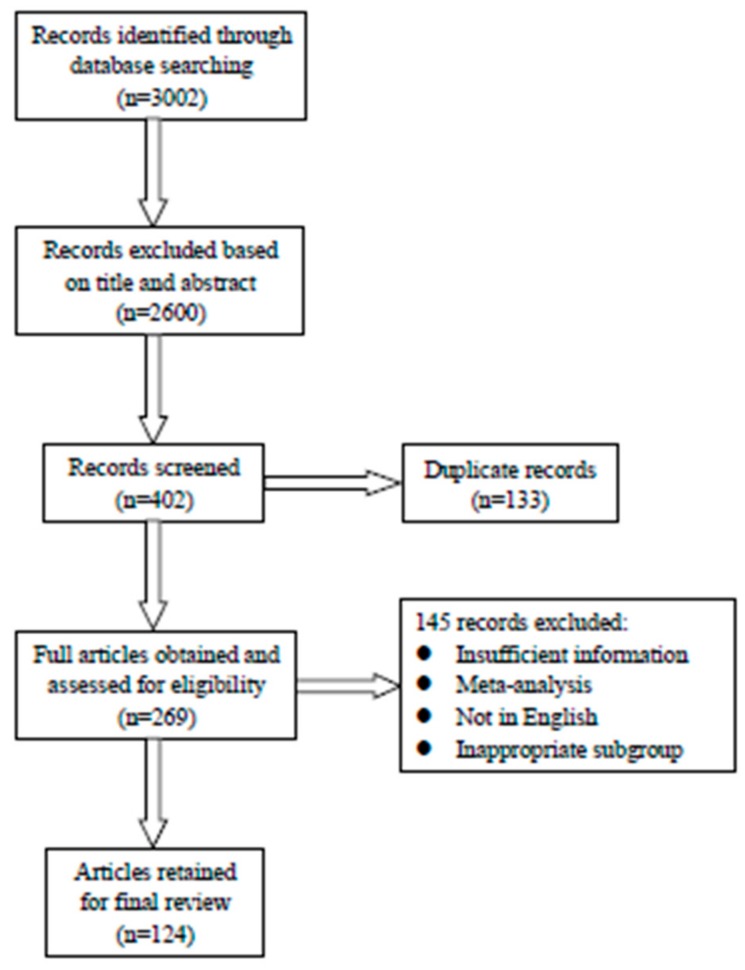
Flow chart of article searching procedures.

**Table 1 ijerph-17-01686-t001:** Comparison of mumps incidence between the pre-vaccine and post-vaccine eras among some countries.

	Pre-vaccine		Postvaccine	
Country [reference]	Years	Annual incidence (per 100,000)	Years	Annual incidence (per 100,000)
USA [6]	1967	100	1993	<0.1
Denmark [4]	1979	726	1995	1
Finland [11,12]	1982	43	1995	<0.1
Slovenia [4]	1979	410	1995	4
Croatia [9]	1985	101	1995	12
France [14]	1986	859	2011	9
England and Wales [10]	1985	40	1995	5
Eastern Germany [13]	1986	155	2016	0.62
Israel [15]	1985	102	1995	10
Thailand [16]	1996	20–70	1997	10–30
Korea [18]	1961	>15,000	1981	<10
Australia [19]	1969	59,000	2002	60
Taiwan [20]	1992	10	2006	1.3

**Table 2 ijerph-17-01686-t002:** Composition of measles-mumps-rubella (MMR) vaccines [72,73,74,75,76].

		Types of vaccines			
Variables	Triviraten	MMR II	Morupar	Priorix	Trimovax
Company	Berna	Merck	Sanofi Aventis	GlaxoSmithKline	Sanofi Pasteur
Measles					
Strain	Edmonston-Zagreb,	Enders Edmonston,	Schwarz,	Schwarz,	Schwarz,
Strength	>1000 TCID_50_	>1000 TCID_50_	>1000 TCID_50_	>1000 TCID_50_	>1000 TCID_50_
Mumps					
Strain	Rubini	Jeryl Lynn	Urabe AM9	RIT 4385	Urabe AM9
Strength	>5000 TCID_50_	>20,000 TCID_50_	>5000 TCID_50_	>5000 TCID_50_	>5000 TCID_50_
Rubella					
Strain	Wistar RA 27/3	Wistar RA 27/3	Wistar RA 27/3	Wistar RA 27/3	Wistar RA 27/3
Strength	>1000 TCID_50_	>1000 TCID_50_	>1000 TCID_50_	>1000 TCID_50_	>1000 TCID_50_

MMR: measles, mumps, rubella; TCID_50_: tissue culture infective dose 50%.

**Table 3 ijerph-17-01686-t003:** Composition of MMRV vaccines [76,78].

	Types of	vaccines
Variables	Priorix-Tetra (GlaxoSmithKline)	ProQuad (Merck)
Route of administration	Subcutaneous	Subcutaneous
Formulation	Lyophilized	Lyophilized
Diluent	Water for injection	Water for injection
Excipients	Amino acids, lactose (anhydrous), mannitol and sorbitol as stabilizers	Sorbitol, sodium phosphate, sucrose, sodium chloride, hydrolyzed gelatin, recombinant human albumin, fetal bovine serum, and other buffer and medium ingredients
Measles		
Strain	Schwarz	Enders’ Edmonston
Strength	>10^3.0^ TCID_50_	>10^3.0^ TCID_50_
Mumps		
Strain	Jeryl Lynn RIT 4385,	Jeryl Lynn
Strength	>10^4.4^ TCID_50_	>10^4.3^ TCID_50_
Rubella		
Strain	Wistar RA 27/3	Wistar RA 27/3
Strength	>10^3.0^ TCID_50_	>10^3.0^ TCID_50_
Varicella		
Strain	OKA	OKA/Merck
Strength	>10^3.3^ PFU	>10^4.0^ PFU

TCID_50_: tissue culture infective dose 50%; PFU: plaque-forming units.

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
