# Peer review of "Current Status of Mumps Virus Infection: Epidemiology, Pathogenesis, and Vaccine"

_ijerph, 2020, doi:10.3390/ijerph17051686_

Round 1
Reviewer 1 Report
Drafted a review Shin-Bin Su et al., on mumps virus infection in a timely manner when developed countries are reporting sporadic outbreaks of mumps virus infection. The authors have drafted a very concise and important review for a broad audience. To further improve the quality of the review, the reviewer strongly suggests to add a section, ‘Emerging Mumps Virus Infection’ and discussed possible reasons and mitigation steps in this section.
Author Response
Thank you for your comments. We have corrected our manuscript point by point as attached. Please check them. Thank.

Reviewer 2 Report
This reviewer commends the authors for their research titled, “Current status of mumps virus infection: epidemiology, pathogenesis, and vaccine development.”
Questions and comments to the authors:
Background
- The background is characterized by a paucity of evidence-based information.
- The authors need to address the rationale and the objectives for doing a systematic review of mumps of all vaccine-preventable diseases.
- The review needs to include and elaborate public health concerns such as vaccine hesitancy as the important challenges to improving immunization coverage and cause for the reemergence of mumps outbreaks.
- It is also important to include the responses of global health authorities such as WHO and CDC such outbreaks. For example, in response to numerous mumps outbreaks reported throughout the United States in 2016 and 2017, the Advisory Committee on Immunization Practices (ACIP) recommended a third dose of measles, mumps, and rubella (MMR) vaccine for groups of persons determined by public health authorities to be at increased risk for acquiring mumps because of an outbreak (CDC, 2019)
- It is not clear which audience the review is attempting to provide information for: medical community, public health practitioners or the public?
- The term vaccine development that appears in the title presupposes that there is currently the production of new vaccines in some stage of development. According to the CDC, there are six stages of vaccine development: exploratory, pre-clinical, clinical development, regulatory review, and approval, manufacturing and quality control. If the authors have such evidence, they have to include it in the manuscript.
Method
- As is the standard practice in medical and public health research involving systematic reviews, this review recommends that the authors use PRISMA(Preferred Reporting Items for Systematic Reviews and Meta-Analyses)is an evidence-based minimum set of items for reporting in systematic reviews and meta-analyses (Liberati, et. Al, 2009).
- This reviewer also recommends the use of more database combinations for literature searches in systematic reviews: For example, Embase, Web of Science, and Google Scholar in addition to Medline and PUBMED to guarantee adequate and efficient coverage (Bramer, Rethlefsen, Kleijnen & Franco, 2017).)
Results and Discussion
In addition to a historical narrative, authors should organize cutting edge current research evidence around the globe and present it in the manuscript so that policymakers, public health practitioners and the community at large have a new body of knowledge that can be put to use for more effective and efficient public health interventions.
References
Bramer, W. M., Rethlefsen, M. L., Kleijnen, J., & Franco, O. H. (2017). Optimal database combinations for literature searches in systematic reviews: a prospective exploratory study. Systematic reviews, 6(1), 245. https://doi.org/10.1186/s13643-017-0644-y
CDC. (2019). Using a 3rd Dose of MMR. Retrieved fromhttps://journals.lww.com/jphmp/Abstract/2020/03000/CDC_Guidance_for_Use_of_a_Third_Dose_of_MMR.3.aspx
Liberati, A., Altman, D. G., Tetzlaff, J., Mulrow, C., Gøtzsche, P. C., Ioannidis, J. P., ... & Moher, D. (2009). The PRISMA statement for reporting systematic reviews and meta-analyses of studies that evaluate health care interventions: explanation and elaboration. Annals of internal medicine, 151(4), W-65.
Author Response

(The authors gave the same response as above.)

Round 2
Reviewer 2 Report
I commend the authors for their point to point response to most of this reviewer's concerns and comments. However, the PRISMA Flow diagram is missing. I strongly recommend the authors incorporate their article-selection process using the attached flow diagram.

Author Response
Thank you for your comments. We added the flow chart of article searching process in Text (line 64). Thank again.